# Using the Capability Approach to Review the National Legislative Frameworks for Support Services for Persons with Disabilities in Four Countries in Asia

**Shivani Gupta** [1,*]**, Agnes Meershoek** [2] **and Luc P. de Witte** [3]

1   CBM Christoffel-Blindenmission Christian Blind Mission e.V. Stubenwald-Allee 5, 64625 Bensheim, Germany
2   School CAPHRI, Care and Public Health Research Institute, University of Maastricht,
    6200 MD Maastricht, The Netherlands
3   Centre of Expertise Health Innovation, The Hague University of Applied Sciences,
    2521 EN The Hague, The Netherlands
*   Correspondence: shivani.gupta@cbm.org

**Abstract:** Implementation of the United Nations Convention on the rights of persons with disabilities (UN CRPD) requires countries to harmonise their legislative frameworks with it. This paper investigates the national legislative frameworks of four Asian countries to see the extent to which they provide support services in accordance with Article 19 of the UN CRPD. The UN CRPD requires persons with disabilities to have *access to* and *choice and control* over support services. To analyse the policy alignment with the UN CRPD, an analytical framework based on the Capability Approach (CA) was developed. The results show that most countries address support services, including assistive devices, only from the perspective of a social security measure for persons with disabilities living in poverty, failing to uphold the rights of those not meeting those eligibility criteria. However, while support services are inseparably linked to social security, they also are a right for persons with disabilities. Therefore, a paradigm shift is required in the approach of support services and the distributive systems of countries, from one that addresses persons with disabilities as those *requiring care* considered a *burden*, to one that considers them rights holders with equal opportunities, for which, support services are a pre-requisite.

**Keywords:** capability approach; support services; choice and control; eligibility; role of the family; agency; disability

## 1. Introduction

Persons with disabilities are known to face challenges in leading meaningful lives because of a lack of adequate and targeted support that enables them to exercise their human right to self-determination and equality. Several international agencies have confirmed that the provision of appropriate support is a key enabler for a large number of persons with disabilities who are struggling to live self-determined lives, but the need for support for most remains largely unmet [1,2]. To ensure countries uphold their rights, the United Nations adopted the Convention on the Rights of Persons with Disabilities (UN CRPD), which is a legally binding human rights treaty that came into force in 2007. Therefore, States that ratify it are obliged to implement and uphold these rights in their countries. Article 19 of the UN CRPD calls upon the States to ensure that all persons with disabilities have equal right to live independently and participate effectively in their communities. To realise this right, it stresses on the need for them to access a range of support services in their community [2]. Article 28 links support services to social protection and requiring countries to make these available to persons with disabilities [3].

There is no specific definition of support services. According to the special rapporteur on the rights of persons with disabilities, "support is the act of providing help or assistance



to someone who requires it to carry out daily activities and participate in society" [2] (p. 5). For persons with disabilities, it specifically includes a wide range of interventions that may be offered formally by paid service providers, or informally, primarily by their families and personal networks. It includes a range of options from personal assistance to community living support, communication support, respite care services, information and decision-making support, assistive devices and so forth [2,4,5]. Formal support mechanisms are most often paid for by the government through public money and provided by contracting private service providers. Often a multi-disciplinary team including occupational therapists evaluates and suggest the support services required by an individual in a formal setup. However, globally, informal support provided voluntarily by family and friends is most prevalent [4]. States at times in the absence of formal support systems may also pay informal support providers through mechanisms such as caregivers' allowances.

The UN CRPD itself does not prescribe any model for providing support services, but provides overarching principles for delivering them, stating that all persons with disabilities must have "equal access to, and equal choice of, and control over support services that respect their inherent dignity and individual autonomy and aim to achieve effective participation and inclusion in society" [5] (p. 10). Thus, the overarching principles look at *access* and *choice and control* as key elements for providing support services. To assess how different countries address these key principles, this paper investigates the question: "How successfully have the national legislative frameworks for support services operationalised the UN CRPD?" The focus of this research is on personal assistance services required for persons with disabilities for activities of daily living and mobility. The research looks at four countries in Asia, namely India, Nepal, the Philippines, and the Republic of Korea (South Korea).

*Disability Theory*

To operationalise the rights-based principles of the UN CRPD, countries need to adopt these into their national legislative frameworks. Implementing social policies relating to support services have a strong link to how countries define and address disability [6–8]. Disability defined in the medical model focuses more on the impairment and interventions are more focused on fixing the impairment which is seen as a problem. Therefore, social policies designed in the medical model will be more attuned towards interventions related to medical and rehabilitation care and eligibility may focus on the extent of impairment without considering aspects like gender, environment, and social differences people live in [6]. The social model in contrast considers disability not as a personal attribute but as a social construct and thus, does not consider the impairment but the barriers in society that handicap a person with disabilities [9]. Social policies based on the social model focus on removing barriers to the participation of persons with disabilities [6,9].

The International Classification of Functioning (ICF) presented by the World Health Organisation (WHO) brings together the medical and the social models of disability in areas of health by considering a biological, individual and social perspective [6,10]. Disability according to ICF begins with the health condition of the person that results in an impairment followed by functional limitations and restriction in participation in a given context. The ICF is designed to be a model and classification system for assessing functioning of a person with disabilities [11]. Functioning covers body functions and structures, activities, and participation, and disability includes impairments, activity limitations, and participation restrictions [6,10]. The ICF is mostly used for normative applications and does not address aspects of equality and justice [11].

This research uses the capability approach (CA) as the analytical framework as it encompasses and also goes beyond the existing models of disability. It links closely to the ICF as functioning is integral to the CA, but additionally looks at the impact of the individuals' functions and the choices they have in using their functioning in a manner that allows them to lead a self-determined life, thus offering a political–theoretical account of equality and justice [6,11]. The CA changes the focus of policy to one that enhances

the functioning and capabilities of the individual to achieve freedom to lead a life of one's choice [8]. It goes beyond the medical model, by considering an impairment as one of the aspects of a human diversity [6,11]. It also goes beyond the social model by focusing not only on external barriers but considering both external and internal barriers individuals with disabilities experience and the opportunities that they have to overcome these barriers [6,8].

The analytical framework used influences the manner in which social justice is defined, including the distributive mechanism of the country that determines how inequality is seen and compensated [12]. In this regard, the CA opposes the existing welfare approach, which most of the reviewed countries follow by looking at social equality in terms of *primary goods*, such as resources, income and wealth. Sen (2009) suggests that *these* are not the appropriate indicators to consider equality when comparing the quality of life or standard of living. According to him, considering income or wealth as indicators falls short of addressing human diversity by ignoring that some individuals, especially those with disabilities, may require more resources to attain the same standard of living [12]. In contrast, Sen (1999, 2009) and other proponents of the CA propose an alternative way of looking at equality, which is based on the *opportunities* and *freedom* individuals have, to choose from and lead a life they value [12,13].

The CA is considered appropriate for this study because it regards the support available to individuals with disabilities as being important for them to benefit equally from the opportunities available, and thus looks at the provision of support as a matter of justice contributing to equalisation of opportunities [14]. Specific to this paper, elaborating on *access*, *choice* and *control* of support services, we theorise *policy eligibility* to define the aspect of *access* to support services. The notion of *agency* is used for examining the *choice* and *control* that support service policies offer. Additionally, the role of the family in the delivery of support services influences *choice* and *control,* and has also been theorised using CA in the next section.

## 2. Materials and Methods

### 2.1. The Analytical Framework

The CA is a broad, normative framework used to assess individual well-being and social arrangements. It has been used extensively in a wide range of fields including social policy [8,15,16]. Sen (2009), elaborating on the equality of opportunities that the CA aims to achieve, defines *conversion handicaps* that are especially encountered by persons with disabilities due to functional limitations to convert the income available into real opportunities, thus requiring support to overcome these [13]. He also suggests that persons with disabilities may require additional resources to overcome these *conversion handicaps* and experience life on an equal basis with others. As a result of these *conversion handicaps* equality in income may yield different lifestyle outcomes for persons with and without disabilities [12]. Such *conversion handicaps* are not addressed by the welfare approach, as it does not look at individual differences in the way people use their income [12].

In the CA, the focus is on the opportunities individuals have to lead a life they value, rather than on income as a gauge for well-being. Opportunities individuals have can be evaluated at two levels—at the level of *functioning* and at the level of *capabilities* [6,12]. *Functioning* are the opportunities a person has to achieve the state of being (such as being well-nourished or cared for) or doing (such as working or learning) and establishes their well-being. *Capabilities* are real opportunities or the freedom of choice a person has to decide which of the various functioning to achieve. Their *capability set* comprises of all the functioning they have the freedom to choose from [6,12,15]. Thus, it establishes the individual's choice and control. This flexibility of evaluating at different levels has been used by disability scholars in policy evaluation [8,16]. Scholars appreciate that the approach considers the agency of the individual, as being central, which implies that persons with disabilities must be given the power and the opportunity to lead their life as they desire [12,17]. Thus, the CA looks at societal measures and the environmental

set-up that empowers individuals to have equal opportunities to live a life they value, regardless of their abilities [12,18,19]. Support services are one such measure required by some persons with disabilities to have equal opportunities.

### 2.1.1. Policy Eligibility

The CA does not propose a definite way of looking at policy eligibility. Therefore, we have explored the definition of disability and other criteria that are commonly considered for eligibility. The CA, although used extensively by disability scholars, does not have a specific definition of disability. However, scholars such as Terzi (2009) and Mitra (2006) have defined ways to understand disability based on the CA [6,14]. Terzi (2009) emphasises that in the CA, social policy must aim to eliminate inequalities in capabilities amongst people [14]. This requires making interpersonal comparisons of capability that recognise human diversity, which means looking at differences in people's personal and external situations and differences in ability to convert resources into capabilities [8,12,14]. Impairment in the CA is considered a personal feature restricting the functioning of the individual, which may result in disability in their interaction with their physical, economic, social and cultural environment [14]. Thus, disability is considered as just one aspect of human diversity, comparable to age and gender [14]. A broad consideration that addresses personal, social and environmental factors when considering the individual's lifecycle is important, as these situations are constantly changing.

### 2.1.2. Agency

The concept of agency is at the heart of the CA and it represents the freedom of choice a person has to choose from all the practical opportunities available to them based on what they value [8,12]. The CA considers individuals to be active participants in their personal and community life rather than being passive spectators or recipients of welfare. According to Sen (1999), the opportunities people have to improve their lives, requires expansion of human agency and freedom, both as an end in itself and as a means of further expansion of freedom [13]. According to Sen (1999) what people can actually achieve depends on their external factors such as their social, economic and political environments, and technological and social modernisation can influence the agency an individual can exercise [13].

Persons with disabilities attaining freedom and involvement in their personal and community life would often require them to have access to support services to achieve their capabilities. First and foremost, policies addressing support services must address such services as an enabler for persons with disabilities to have agency to lead a self-determined life. Additionally, support service policies must look at persons with disabilities having agency as an end in itself. They can do this by offering individuals a range of support options with the freedom to decide what best suits their requirements [20]. Finally, looking at expanding the agency of persons with disabilities as a means would require policies to actively involve persons with disabilities in all processes of getting the support, including evaluation and granting of the service.

### 2.1.3. Role of the Family

The CA does not look at a person in isolation, but as a part of a larger social and environmental structure, of which family is the first unit that can either support persons with disabilities to overcome their conversion handicaps or act as a restraint [8]. According to Sen (2009) *conversion handicaps* experienced by persons with disabilities result in members of their family having to share their capabilities with the family member with disabilities [8,12,21]. For instance, if a person with disabilities is unable to walk, they may not be mobile, unless they receive a mobility aid such as a wheelchair. However, even if they received a wheelchair, they may still not be able to move around because of reasons such as inaccessibility, lack of strength to propel the chair, and so on, and thus would need to depend on the members of the family to push their wheelchair. Sharing of resources by the family can increase

the agency of persons with disabilities, but at the same time, may place their families at a disadvantage as compared to families without a member with disabilities [8,16].

Additionally, the CA recognises that persons with disabilities have a double disadvantage: they earn less income compared to persons without disabilities on account of the personal, social and environmental barriers, and second, they (and their families) incur extra costs of living with a disability, from expenses such as medicines, diet, assistive devices, accessible accommodation and transport. [6,8,12,21]. Therefore, the CA suggests that disability-related extra expenses must be considered when looking at family income as an indicator of well-being [12,21].

*2.2. Methodology*

2.2.1. Country Selection Criteria

This paper reviews the primary legislative frameworks relating to support services of four Asian countries (India, Nepal, the Philippines and South Korea) to understand how they operationalise the CRPD. The CA has been used as the framework for the review. The four selected countries are used as case studies to understand the general situation in the region. These countries have not only different support-service policies but also different economic, demographic and geographical characteristics. They have all ratified the CRPD and therefore have agreed to comply with the principles and standards set by it. In addition, all four countries have been reviewed under Article 33 of the convention on the status of implementation of the CRPD. Three of the four countries have post-ratification legislation and the fourth country has amendments made to the legislation, post-ratification. This indicates that they have started the process of harmonisation. To ensure macro-level regional similarities, only Asian countries were selected and they all concur with the regional instruments for the protection of the rights of persons with disabilities, such as the Incheon Strategy for Persons with Disabilities in Asia and the Pacific [22].

2.2.2. Documents Reviewed

The four countries have diverse legislative documents that address different kinds of support services for persons with disabilities. These documents include legislative acts, policies and rules. The research is limited to investigating the primary legislative frameworks, as legal harmonisation of the primary legislation is one of the first steps towards operationalising the CRPD. Moreover, the legislative framework is the starting point from which other policies and implementation mechanisms evolve. Verification of the data collected from these documents was undertaken by the information and data provided by the state reports submitted to the CRPD committee by the country governments.

The legislations for India, Nepal and the Philippines were downloaded from country government websites. The legislation for South Korea was downloaded from a WHO database. All the legislation was in English and no translation was undertaken by us.

**India**
Rights of Persons with Disabilities Act, 2016 (RPDA) [23]
The National Policy for Persons with Disabilities, 2006 [24]
**Nepal**
The Act Relating to Rights of Persons with Disabilities, 2074 (2017) (ARRPD) [25]
**The Philippines**
Republic Act 7277, 1991 [26]
Republic Act 10754, 2015 [27]
Implementing rules and regulations for RA 7277 and RA 10754 [28]
**South Korea**
Welfare Law for Persons with Disabilities Act, 2013 (WLPD) [29]
Act on Activity Assistance Services for Persons with Disabilities, 2011 [30]
National Policy for Persons with Disabilities 2013 [31]

### 2.2.3. Document Review Process

The review of the legislative documents used an analytical framework developed using the CA. The analytical framework presented in Table 1 elaborates on three factors: eligibility, agency, and the role of the family, with indicators to analyse each of these. Outcomes of the document examination were validated through a comparison with the state report to the CRPD committee.

**Table 1.** Indicators for the review of the primary legislation derived from the CA.

| Capability Approach | Indicators |
|---|---|
| *Access = Eligibility* | |
| Looks at disability as a deprivation that results from impairment and impacts the ability of a person to convert resources into capabilities. | Does the definition of disability make a distinction between disability and impairment? |
| Deprivation faced by persons with disabilities takes into account: (a) personal factors (b) social factors (c) environment factors | Eligibility criteria in addition to disability consider other personal, social, and environmental factors. |
| Deprivation is considered across various parameters acknowledging changes in the person's personal, social and environmental factors at different stages of life. | Does the evaluation process take into consideration of internal and external factors through the life cycle approach? |
| *Choice and control = Agency* | |
| Support services foster the active participation of people in their personal and community lives. | Does the purpose of the legislative framework reflect on the active participation of persons with disabilities? |
| Persons with disabilities can exercise choice and decide on the support they think is best for them. | Is there a range of services offered and is there flexibility in the way these are provided? |
| Persons with disabilities have agency and are active participants in deciding the life they want to live | Does the support service evaluation process actively involve the beneficiaries in the decision-making process? |
| *Choice and control = Role of the family* | |
| Family members are considered an important resource and they agree to share their capability set with persons with disabilities to enable them to have basic capabilities. | Do the support service policies address the additional efforts of the family in supporting a disabled family member? |
| Evaluation of family income takes into account all extra expenses related to disability. | Is there consideration of disability-related extra costs? |

There are three indicators used to investigate eligibility: first, how each country defines disability; second, whether any additional personal, social and environmental criteria are considered in eligibility assessments; and third, the evaluative processes adopted and whether only medical factors are considered, or functional and social factors are also taken into account. Three indicators are used to investigate agency: first, the purpose of the legislative act in question is considered to see if it focused on the agency of persons with disabilities; second, the range of support services offered are investigated to see if they allowed persons with disabilities to exercise choice; and finally, the process of evaluating additional criteria for support services are examined to see if their opinions and their families are taken into account.

The role of the family is looked at, using two criteria: the first indicator involves an analysis of the role legislation assigned to the family and whether it accounted for the extra efforts of the family for supporting a family member with disabilities; the second indicator

examines whether the legislation addressed disability-related extra costs incurred by the person with disabilities and their family.

## 3. Results

The following section provides a review of the primary legislation for support services for persons with disabilities in the four countries, based on the indicators presented in Table 1.

### 3.1. Eligibility

First, the four-country legislations reviewed adopt similar approaches to describing a person with disabilities. The disability acts in India and the Philippines have adopted the definition from the convention and have the same definitions of a person with disabilities—"person with long-term physical, mental, intellectual or sensory impairment which, in interaction with barriers, hinders his full and effective participation in society equally with others" [23] (p. 3); [28] (p. 1). In Nepal, the Act Relating to the Rights of Persons with Disabilities, 2074, 2017 defines a person with disability as "a person who has long-term physical, mental, intellectual or sensory disability or functional impairments or existing barriers that may hinder his or her full and effective participation in social life on an equal basis with others" [25] (p. 1). For implementing this broad definition of disability, India, Nepal and the Philippines provide a more medical definition or list of the impairments that supplement the definition. South Korea defines persons with disabilities as "those who are considerably restricted in their daily and social life for a long period of time due to their physical or mental disabilities" [29] (p. 1) and categorises disability into 15 different impairments [31].

Policies in all four countries apply an additional layer of qualifying criteria for eligibility for support services. In India, according to the RPDA 2016, persons requiring *high support* are eligible. *High support* is defined as "an intensive support, physical, psychological and otherwise, which may be required by a person with benchmark disability for daily activities, to take independent and informed decisions to access facilities and participating in all areas of life including education, employment, family and community life and treatment and therapy" [23] (p. 3). Benchmark disability considers "persons suffering from not less than 40% disability" [23] (p. 3). In Nepal, the ARRPD, 2017 also defines *helpless people with disabilities* who are eligible for government support as "a person with disability who does not have any property or family member or guardian to attend, care and serve him or her or who cannot earn their living by way of self-employment" [25] (p. 1). The Philippines provides support to *marginalised disabled persons*, who are defined as "persons who lack access to rehabilitative services and opportunities to be able to participate fully in socio-economic activities and who have no means of livelihood or whose incomes fall below the poverty threshold" [26] (p. 3). In South Korea, only "persons with serious disabilities of a degree equal to or more severe than the degree of disability prescribed by Presidential Decree, who have difficulty in leading daily and social lives by themselves" [30] (p. 2) are considered eligible to access support services. Other than the severity of the disability, South Korea also considers the standard of living of the family and levels of financial support. While calculating the cost of supports to the person with disabilities, the family's standard of living is considered.

All countries require a certificate or an identity card verifying a person with disabilities, which is a primary requirement to access support services offered by the government. In India, a disability certificate can be obtained from a certifying authority competent to issue it. The Gazette of India: Extraordinary Part II—Sec.3 (ii) issued in 2018 provides the guidelines for the assessment of various disabilities under the RPDA, 2016. It requires the person with disabilities to undergo a clinical examination by a medical doctor. In Nepal the certificate is given based on four classes: profound, severe, moderate and mild [14]. These classes are defined based on the functioning of the person to undertake daily activities and participate in social activities. The certificate can be obtained from the local level ward

officer who issues it if the disability is visible, and only where the disability is not visible, is a medical examination required [25]. The Philippines depends on the Department of Health to evaluate persons with disabilities and a disability identification card is issued only after undergoing a medical evaluation [32]. In South Korea, the severity of a person's disabilities is graded from 1 to 6 degrees of impairment through medical evaluation.

To summarise, the eligibility mechanisms, all countries have two levels of evaluation: first, identifying who is legally a person with disabilities, since the services are offered through the government-aided system are only available to them. The definition of a person with disabilities in the legislative frameworks are broad, considering not just the impairment but also other internal and external factors that influence the functioning of the individual. However, the evaluation processes continue to look at the severity and type of impairment with a medical evaluation process that focuses solely on the impairment and requires a medical practitioner to undertake it. Therefore, the first level of evaluation is only concerned with the medical diagnosis, and the multi-dimensional definition of disability that is adopted in the law does not make a significant impact on the ground. Second, an extra definition is adopted by the four countries, potentially expanding the scope of the eligibility criteria for accessing support services. For instance, the definition of *high support needs* in India expands the eligibility criteria to consider social and environmental factors. In the Philippines, the definition of *marginalised persons with disabilities* considers the presence of a family as an additional criterion. The definition of *helpless persons with disabilities* from Nepal focuses on income, while South Korea considers the severity of impairment and the family's standard of living as an additional criterion. The additional definitions adopted, however, do not address the multidimensional factors that individuals encounter.

### 3.2. Agency

The first indicator suggests that, except for South Korea, none of the four countries have drafted specific legislation concerning support services, but rely on sections of general disability acts when addressing support services. Hence, support services fall under a broader scope of the act, as provision of rights and benefits at large. In India, Chapter 7 of the RPDA 2016 Act explains the process for providing support to persons with *high support needs* so that they can live and participate in all spheres of life [23]. In Nepal, the purpose of the ARRPD 2017 is to uphold the rights of persons with disabilities to live in the community and elaborates "the persons with disabilities shall have the right to obtain assistive materials and community assistance in order to earn the living respectfully" [25] (p.6). The RA 7277 of the Philippines is also a general disability act. Chapter 4 of this act focuses on *auxiliary social services* and describes the different types of support that can be offered. It is aimed at "ensuring that marginalised persons are provided with the necessary auxiliary services that will restore their social functioning and participation in community affairs" [26] (p. 8). Similar to the legislation drafted in South Korea, the range of *auxiliary social services* prescribed in the act are aimed at community participation and on maximising the social functions of persons with disabilities. The purpose of South Korea's Act on Activity Assistance Services for Persons with Disabilities is to "raise the quality of life of persons with disabilities by assisting persons with disabilities to live with self-reliance and lift the burden on their families through providing activity support allowance for persons with disabilities" [30] (2011, p. 1).

The range and options of support services available that directly impact the choices that persons with disabilities have to lead self-determined lives [26]. However, most countries have very limited options in terms of the support offered. In India, the act does not specify any particular type of support that should be offered, but leaves it open, stating that an eligible person may receive support "subject to relevant schemes and orders of the appropriate Government in this behalf" [23] (p. 14). According to the act, access to support is subject to relevant schemes, without any certainty in terms of when, what and how support services will be provided. In Nepal, the ARRPD 2017 also does not provide a list of the services that may be offered to persons with disabilities [25]. In the Philippines

there is a wider range of services that are offered, including the provision of prosthetic devices, communication skills training, mobility training, and training in enhancing daily living capabilities [26]. Most of the services offered are rehabilitative and may not be available in the long term. In South Korea, the act offers a wide range of support services, including bathing, nursing, and night-time support. Support may also be provided in their home. The person with disabilities receives a coupon from the authorities to buy the support required; these coupons can be used to obtain services from different government-recognised, service-providing organisations [30], giving some choice and control to the person with disabilities in terms of who supports them. However, there is a top limit to the hours of support available in a month, after which they have to pay the full cost of support or manage unsupported.

In India, the act states that persons requiring *high support* must request assistance by making an application to an authority who should then forward this request to an assessment board who decides whether the need for support is genuine, and the form and nature in which it is delivered [23]. In Nepal, government support is available to *homeless persons with disabilities* and there is no evaluation process elaborated, except by visual confirmation [25]. In the Philippines, all services, except for substitute family care, are provided at the municipal and city government levels. The responsibility for providing services lies with the mayor, who appoints a social worker from the Department of Social Welfare and Development for effective implementation. The social worker and their team work closely with persons with disabilities and their families in the community [26]. In South Korea, to access support services, persons seeking it have to make an application, which is evaluated by the Entitlement Deliberation Committee [30]. The persons with disabilities or their families are not a part of the committee, and therefore do not have a say.

To summarise, the level of agency that persons with disabilities are able to exercise, is closely related to the eligibility criteria and the evaluation process for getting support services. Of the four countries reviewed, only South Korea has an act that addresses support services, while the rest address support services as a part of the main disability rights legislation. Having a separate Act has an impact on the amount of support available to persons with disabilities and the way it is provided, which increases agency of persons with disabilities. As was evident from the secondary level definition of eligibility, countries that look at support services as a part of the main disability legislation, consider them only as a social security measure. However, in South Korea the separate act addressing support services considered support services as a pre-requisite for all persons with disabilities. They offer a greater range of services, and the buying power to the persons with disabilities through vouchers. This increases the choice and control they have over the support services they access. However, even in South Korea the evaluation to decide which support to grant does not include persons with disabilities, thereby reducing the control they have.

### 3.3. Role of the Family

The support offered in the four countries is closely linked to the family of the person with disabilities, regardless of their age. This was investigated using two indicators. First, for those with families, the family income or the standard of living is important when eligibility for support is being assessed. Those without a family are most likely to be supported in an institution-like set-up. In India, the RPDA 2016 does not hold the family responsible for person with disabilities. However, it recognises the importance of families in terms of the provision of care and support. This act also requires creation of facilities for those without families. In Nepal, the ARRPD 2017 puts a specific legal obligation on the family to provide support in education and for hospital visits. It also requires the local government to impart training to the family members in providing care to the family member with disabilities. Similarly, South Korea identifies family members as "person(s) obligated to support persons with disabilities" [30] (p. 1). These countries thus impose a legal duty on the family. However, in the Philippines and South Korea, the family, together with the government and NGOs, shares the responsibility of ensuring that persons with

disabilities can live as independently as possible. They do this by providing support in the community [26,33].

In the Philippines, the range of services provided, addresses family needs such as developing the capacity of the family to support the family member with disabilities and providing after-care services that enable persons with disabilities to settle into their family and community after returning home from institutional care [26]. South Korean legislation illustrates the need for the governments and agencies involved in providing care and support to lift the *burden* of care from the family. The act requires "lifting the burden on their families through prescribing matters concerning an activity support allowance for persons with disabilities" [30] (p. 1). Adults with disabilities should be able to build lives of their own, where they have the choice of whether they would like to live independently or be with their families.

The second indicator looks at whether the extra expenses related to disability are addressed by the legislative document. Living with a disability may place an additional economic demand on persons with disabilities and their families. Therefore, examining the impact of such disability-related extra costs is important. All four countries reviewed recognise that there are extra costs related to disability, but they address these in different ways. The act in India makes provisions for paying a caregiver allowance to the person with *high support needs*. This is a social security measure that can also be given to family caregivers. The National Policy for Persons with Disabilities (2006) in India also recognises that persons with disabilities, their families, and caregivers may "incur substantial additional expenditure for facilitating activities of daily living, medical care, transportation, assistive devices, etc. . . . " [24] (p. 7). It discusses the possibility of offering tax exemptions to them and their family members, and it also makes provision for a disability pension. The ARRDP 2017 in Nepal offers no support to the family, except for the training mentioned earlier. In the Philippines, a 20 per cent discount is offered to persons with disabilities on many public services. Persons caring for and living with them are offered Income Tax exemptions [27]. In South Korea, public enterprises are required to "endeavour to lighten the economic burden of the disabled and other persons supporting such disabled persons" by cutting taxes and reducing fees for public facilities. The AAASPWD, 2011 requires the government to raise funds on an annual basis to assist persons with disabilities to be more self-reliant, and to reduce the pressure of care and support on the families [30].

In summary, the legislative frameworks look at the family as being responsible for the well-being of the adult family member with disabilities. As a result of not having an option, such an arrangement can not only encumber the family but also compromise on the agency of both the adults with disabilities and their family members. In South Korea, although a wide range of services are provided by the government, the family is supposed to be responsible, irrespective of their age or the age of the family member with disabilities. Putting an obligation on the community as is done in the Philippines may foster their inclusion in community life, as the community at large becomes responsible for the well-being and inclusion of persons with disabilities. Moreover, while the four countries recognise disability-related extra costs, these are considered as pertaining to the family and not of the person with disabilities. That leaves them largely dependent on their family, making it difficult for them to become independent or self-reliant.

## 4. Discussion

This research looks at how the legislative frameworks are operationalising support services for persons with disabilities using a framework based on the CA. The research was limited to investigating the primary legislations in India, the Philippines, Nepal and South Korea. The investigation revealed a variation in the level of compliance and varying degrees of success in terms of ensuring access, choice and control. While none of the four countries fully comply with all the indicators used, there are elements in compliance that may be seen as good examples. For instance, the definition of disability in all countries is according to the convention; in South Korea, support services are recognised as a pre-requisite for

participation, and a specific act has been enforced to provide these. Moreover, not all countries consider the family to assume the legal responsibility for the adult family member with disabilities. Some countries also recognise the responsibility of the community in ensuring inclusion of persons with disabilities.

Of the four countries the support services legislation in South Korea seems most compliant with the UN CRPD. A simplistic answer for this difference may be that they are a high-income country and therefore have more funding to provide support services, but investigating deeper highlights other factors. To begin, the purpose of the act on support services is to enable persons with disabilities to live more independently which is not so in the other countries. As a result, in South Korea support services are seen as a right for all persons with *severe* disabilities to have access to and not limited to those living in poverty or without a family like in the other three countries. They use a sliding scale based on family income to decide the share of the support service cost that eligible persons with disabilities must contribute, thus making everyone with a certain severity of disability eligible. A benefit of not poverty-targeting support services can be a higher demand for them that allows countries like South Korea to offer a wide range of services to meet individual requirements.

The UN CRPD proposes that access to support services is a matter of right and it cannot be limited to being a social security measure available to only certain persons with disabilities [1], which is reiterated by the CA Terzi (2009) [14]. The CA, suggests looking at the personal characteristics of the individual to convert resources into valuable ends when deciding the support service offered to individuals. [12,14]. A limitation of considering support services only for those living below the poverty threshold like in India, Nepal and the Philippines, is that it recognises only some selected needs of persons with disabilities and offers predefined solutions based on this understanding [7]. This conflicts with the need to develop support that is tailored to individual requirements, an approach that allows persons with disabilities to live with dignity and to exercise autonomy as is required by the CRPD [1]. It also restricts the development of community support services as the demand is for only some predefines services.

The eligibility in most government systems for offering aid focuses on targeting resources towards fixing the impairments, as in the medical model, or removing specific barriers from the environment, as in the social model. These, however, do not take into account the families of persons with disabilities who are in caregiving roles. Alternatively, according to the CA, the distribution system must be based on the equalisation of opportunities and recognise the principle of agency, which implies offering individualised support that enables persons with disabilities to lead self-determined lives and increase the choice and control that they have [8,12,18]. However, changing the country's traditional eligibility policies and practices is difficult. The existing eligibility policies largely focus on compensating for loss of work, whereas strategies that address and compensate the *conversion handicaps* that persons with disabilities encounter while accessing equal opportunities for equal wages is required. This calls for a radical shift in thinking and practice.

Making such intrinsic changes may not be easy for several reasons. First, since disabled persons are a heterogeneous group, their support requirements are diverse, and a one-size-fits-all approach will not work. Second, persons with disabilities and their families are rarely able to afford to pay for formal support services especially when there is no government action to develop, subside or monitor the support services in the community. As a result, there is inadequate demand for it. Third, traditionally formal support systems are not well recognised in most countries and the family is expected to support the family member with a disability as is seen also in the review of the country legislations. Such a traditional way of thinking and lack of spending power to pay for formal support services discourages the development of support services.

Elaborating further, there are multiple outcomes of making the families responsible to provide support to the disabled family member. First, in a more positive light, such a system ensures that persons with disabilities are looked after, especially in countries

where support services are underdeveloped and the economic situation of the country may not make it easy for the government to develop these. Second, though not as positive, by putting all responsibility of care and support on the family, the government is putting an undue financial, physical and social burden on the family, as has been elaborated extensively in other literature [4,21]. Finally, there is research suggesting that mandating families to look after the member with disabilities may make it difficult for both the person with disabilities and their family to have any choice or control of their life [4,34] and also discourage developing alternate support system [7].

Though legislations make families responsible for the family members with disabilities, they have started recognising the importance of the family in supporting the member with a disability to overcome their *conversion handicap* [12]. This is evident as the legislations are addressing training to the families in supporting persons with disabilities, and in some countries, offering a caregiver's allowance that may be useful in providing compensation for extra disability-related costs incurred by these families including the loss of income. Some countries also offer incentives such as social security benefits, pension schemes, concessions, and tax benefits as compensation. While such systems are important in the present set-up for families to have incentives and monetary support to assume the role of primary support providers, in the long run, these do not address the support requirements of the person with disabilities or develop alternate support options in the country [2,5]. Moreover, support for the family cannot to be considered a substitute for the support for the person with disabilities [2].

Another aspect to consider while offering compensation for the family or the person with disabilities is to be mindful of how these resources are distributed within the family and the control persons with disabilities have over it. For instance, according to CA, persons with disabilities may often receive much fewer benefits as compared to non-disabled family members, thus not addressing the differences in the standard of living of families with and without a family member with disabilities [21]. Kelly [35], however, points out that such grants may be the only regular family income in many cases, and therefore, if targeted for persons with disabilities rather than the family, can promote interdependence within the household, with persons with disabilities supporting the family financially in return for care.

These countries continue with the idea that support services are about being cared for, which enables the person with disabilities to survive but does not guarantee any personal autonomy. Traditionally, this role is fulfilled by the family, irrespective of the age. Such an approach reduces persons with disabilities to being passive recipient of *care* or a *burden* on the family, rather than right holders where support is a pre-requisite for equal participation [2]. Such a perspective on the one hand limits the range of support services offered, leaving persons with disabilities with no choice and control. Additionally, on the other hand, since the family is responsible for providing care, the absence of alternate mechanisms leaves them without any choice or control. According to Lang (2009), in developing countries, since ideologically economic and social policies are based on the notion of charity and welfare, bringing radical shifts may raise challenges in making changes in the legislative framework and in the way it is implemented [36]. This is true even when we look at policies for support services where making them compliant with the CRPD remains a challenge, because it is not easy for countries to make radical changes in their existing and established systems which focus on providing *care* and reducing the *burden* of the family. Such perspectives reinforce the existing negative perception of disability, preventing persons with disabilities, their families and the community from demanding change [37]. While support services are linked with social security also in the UN CRPD, such services are a right, and therefore need to be viewed from a wider frame than addressing them only as a targeted social security measure for the poor.

## 5. Contribution, Limitations and Further Research

The unique contribution of this research is in the use of the capability approach framework to theorise support services for persons with disabilities. The capability approach has been used by several disability scholars and this research adds to that existing body of research that considers the CA, a normative framework about social justice and development and about human well-being, well framed to operationalise international human rights laws such as the UNCRPD and the rights of persons with disabilities [6,12,19,20,38]. Furthermore, the CA gives an opportunity to evaluate beyond functionalities, and consider the choice of people based on their agency, which goes beyond the ICF that considers functionalities in the social context of the individual or the social model of disability that focuses on external barriers. Therefore, theorizing support services using the other disability theories would not have given us the same conclusion of the analyses that the CA gave us, which highlights the need for bringing a rights-based paradigm shift in the design, availability and delivery of support services in different countries. The research can be seen as the starting point for more detailed research in the future based on the analytical framework developed by reviewing the primary legislation and the State reports of the government to the committee on the rights of persons with disabilities in the four countries in Asia. In the future, therefore, more in-depth study may be undertaken by looking at the ground implementation of the national and local-level policies with a broader canvas of country-specific literature and outcomes of these policies based on the perspectives of the persons with disabilities and their families. This would not only help validate and finalise the conceptual framework but also address the second limitation of this research, which was the legislative documents reviewed were in English and translated by official government sources; by using these we assume that governments have oriented themselves in the translation of the UN CRPD terminology. In-depth country-level research may provide opportunities to capture the nuances of the local language and culture.

## 6. Conclusions

Support services are a pre-requisite for equal participation of persons with disabilities, and thus having access to such services is a right for all persons with disabilities. However, existing policies addressing support services largely look at them only from the lens of social security that is largely offered to persons below a certain income threshold, thereby ignoring persons with disabilities who may be just above the threshold or those who can pay for the services. Moreover, legislation is required where support services are available to persons only with certain types and severity of disability, discriminating against other disabilities. With support services restricted to being addressed only as a social security mechanism, expecting the family to provide *care* to the family member with disabilities negates the opportunities for persons with disabilities or their family to exercise any choice and control in their lives. It is therefore suggested that to make support services more CRPD-compliant, a paradigm shift is required in the way these services are seen to one that considers support as a service available in the community, rather than being looked at as the providing of care by the family. Having adequate and appropriate support goes beyond *providing care* and *reducing the burden*; it enhances the choice and control persons with disabilities have on their lives. This may require countries to modify their understanding of support services and their social security mechanisms to one that allows all persons with disabilities to exercise their right to live in the community and have capabilities on an equal basis with others.

**Author Contributions:** Conceptualization, S.G., A.M. and L.P.d.W.; methodology, S.G. and A.M.; software, S.G.; validation, S.G., A.M. and L.P.d.W.; formal analysis, S.G.; investigation, S.G.; resources, A.M.; data curation, S.G.; supervision, S.G.; project administration, A.M.; funding acquisition, L.P.d.W. All authors have read and agreed to the published version of the manuscript.

**Funding:** This research was funded by NUFFIC, grant number NFP-PHD CF-11830.

**Institutional Review Board Statement:** Not applicable.

**Informed Consent Statement:** Not applicable.

**Data Availability Statement:** Not applicable.

**Conflicts of Interest:** The authors declare no conflict of interest.

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
