# Peer review of "Using the Capability Approach to Review the National Legislative Frameworks for Support Services for Persons with Disabilities in Four Countries in Asia"

_societies, doi:10.3390/soc12060185_

Round 1

Reviewer 1 Report

This paper focuses on personal assistance services required for persons with disabilities for activities of daily living and mobility. The research looks at four countries in Asia, namely India, Nepal, the Philippines, and South Korea.

Overall:

1.       The article presented a good review to help people with disabilities.

2.       The paper showed well organized with proper structure, and the methodology and results are communicated.

3.       The article showed the limitation of the study, which is a good point.

This paper can be accepted. However, there are some points that the authors can handle:

1.       You have to highlight the novel contribution of the study.

2.       It is better to compare your work with some of the related work. You have to give more details about how your idea is better than theirs.

3.       It would be better to add a section for your future work.

4.       The abbreviation must be written beside the words when it appears for the first time. The authors should check them as (UN CRPD) and (CA) was written twice.

5.       You have to update the references because most of them were before 2011, which is very old.

Author Response

Dear Esteemed Reviewer,

Thank you for your comments which have been very useful in strengthening the manuscript. We have incorporated a number of your comments in track mode in the manuscript and answered the comments in blue below. The ones we have not been able to add, we have provided a justification for not including them.

Thank you again

Best wishes,

(Co-author of the manuscript)

Reviewer 1

This paper focuses on personal assistance services required for persons with disabilities for activities of daily living and mobility. The research looks at four countries in Asia, namely India, Nepal, the Philippines, and South Korea.

Overall:

  1. The article presented a good review to help people with disabilities.
  2. The paper showed well organized with proper structure, and the methodology and results are communicated.
  3. The article showed the limitation of the study, which is a good point.

This paper can be accepted. However, there are some points that the authors can handle:

  1. You have to highlight the novel contribution of the study.

We have added the novel contribution of the study in the sub-section in the discussion titled ‘Contribution, Limitations and Future Research’.

2. It is better to compare your work with some of the related work. You have to give more details about how your idea is better than theirs.

Addressed in the sub-section in the discussion titled ‘Contribution, Limitations and Future Research’.

  1. It would be better to add a section for your future work.

We have addressed this comment in the new sub-section in the discussion titled ‘Contribution, Limitations and Future Research’.

  1. The abbreviation must be written beside the words when it appears for the first time. The authors should check them as (UN CRPD) and (CA)

We have provided those abbreviations twice with the written elaboration – once in the abstract and once in the main manuscript to enable readers to read them independently

  1. You have to update the references because most of them were before 2011, which is very old.

We have largely used three types of references 1Theoretical works that remains relevant over long periods of time; 2. Legislations and policies that may be old but are up to dated. 3. United Nations explanation guidance that again remain relevant over time.

We have however added a couple of newer references

Reviewer 2 Report

Thank you for allowing me to review the manuscript. Although I think it is interesting, I think the work needs a restructuring because it is complicated to follow and the way the results are somewhat tedious, I think that with a better structure the work would benefit from it. Below I leave comments for the authors

Keyword: disability???

Introduction

I don't understand the references because they start at 29

I think that if they include the articles they should indicate the page where said article is within the law

Regarding the “range of interventions that may…..” I think you should include multidisciplinary intervention teams, especially when it comes to activities of daily living, where occupational therapists are the professional with these skills and they are not named in the introduction.

Disability theory

It is great that they include the ICF but they should talk about social determinants of health from the WHO, which would be appropriate given the theme of the manuscript.

“The ICF is designed to be a model and classification system for assessing functioning of a person with disabilities”. The ICF classifies any person... not just disability, it doesn't seem appropriate to me to include this like this. The authors should talk about the core set... and the section on environmental factors in a more specific way where the topic to be addressed in the manuscript is included

Materials and Methods

This section is quite complicated to follow, I do not understand the design or the procedure that has been followed. I do not understand why they classify it as they do having the CIF…. They could follow their different dimensions.

Methodology

Why on the one hand material and methods and then methodology?? Now the revised documents are understood…. It is necessary to structure this article, then on page 6 they explain the indicators... and the material and methods part is not understood. Although I still think that I do not understand these indicators when they have the CIF to structure...

Results

I think that this section needs to be structured, and I also think that the use of figures can improve….

discussion

What is the reason for including the APA standard (WHO, 2001).

Author Response

Dear Esteemed Reviewer,

Thank you for your comments which have been very useful in strengthening the manuscript. We have incorporated a number of your comments in track mode in the manuscript and answered the comments in blue below. The ones we have not been able to add, we have provided a justification for not including them.

Thank you again

Best wishes,

 Co-author of the manuscript

Reviewer 2

Thank you for allowing me to review the manuscript. Although I think it is interesting, I think the work needs a restructuring because it is complicated to follow and the way the results are somewhat tedious, I think that with a better structure the work would benefit from it. Below I leave comments for the authors

Keyword: disability???

We have added disability as a keyword

Introduction

I don't understand the references because they start at 29

We have adjusted the referencing

I think that if they include the articles they should indicate the page where said article is within the law

In the introduction mention of an article is related to the UNCRPD. It is common practice to refer to the article and not the page no of the UNCRPD. Page numbers have been provided later in the text where the exact words are quoted from the country legislations.

Regarding the range of interventions that may…..” I think you should include multidisciplinary intervention teams, especially when it comes to activities of daily living, where occupational therapists are the professional with these skills and they are not named in the introduction.

We have added the recommendation to the manuscript

Disability theory

It is great that they include the ICF but they should talk about social determinants of health from the WHO, which would be appropriate given the theme of the manuscript.

“The ICF is designed to be a model and classification system for assessing functioning of a person with disabilities”. The ICF classifies any person... not just disability, it doesn't seem appropriate to me to include this like this. The authors should talk about the core set... and the section on environmental factors in a more specific way where the topic to be addressed in the manuscript is included

We disagree as the social determinants in the ICF focus more on the health condition. Moreover, since the paper is based on the CA it focuses more on functioning and capabilities as defined in the CA.

Materials and Methods

This section is quite complicated to follow, I do not understand the design or the procedure that has been followed. I do not understand why they classify it as they do having the CIF…. They could follow their different dimensions.

Methodology

Why on the one hand material and methods and then methodology?? Now the revised documents are understood…. It is necessary to structure this article, then on page 6 they explain the indicators... and the material and methods part is not understood. Although I still think that I do not understand these indicators when they have the CIF to structure...

We have clubbed the response for recommendations for the Materials and Methods and the Methodology together. This research is a theoretical piece of work that operationalises the capability approach in the context of support services.    ICF is more efficient for mapping rather than providing a detailed insightful theory. Using the ICF would most likely bring us to a very different conclusion which may or may not be based on the rights-based perspective which we want from this research.

Results

I think that this section needs to be structured, and I also think that the use of figures can improve….

We disagree with the recommendation for restructuring the results as the analysis of each indicator first reviews the country legislation followed by an explanation on what that implies. Moreover, the first  reviewer found structure of the manuscript appropriate. We also feel that adding figures would be difficult especially since it’s a policy review and creating figures in policy reviews are not common.

discussion

What is the reason for including the APA standard (WHO, 2001). Do

We have made adjustments

Round 2

Reviewer 1 Report

The authors improved their manuscript significantly. They were able to respond to my previous question satisfactorily. So, I will accept this paper for publication. However, you have to check the order of the references because now they do not appear in order.

Reviewer 2 Report

Thanks for taking into consideration the comments and suggestions made in the previous review, I think the article is suitable for publication except for a small change in a line that does not correctly include the name of the classification.

Line 79 International Classification of Functioning, Disability and Health (ICF)